
# Constraining the gauge-fixed Lagrangian in minimal Landau gauge

**Axel Maas**

Institute of Physics, NAWI Graz, University of Graz,
Universitätsplatz 5, A-8010 Graz, Austria

## Abstract

A continuum formulation of gauge-fixing resolving the Gribov-Singer ambiguity remains a challenge. Finding a Lagrangian formulation of operational resolutions in numerical lattice calculations, like minimal Landau gauge, would be one possibility. Such a formulation will here be constrained by reconstructing the Dyson-Schwinger equation for which the lattice minimal-Landau-gauge ghost propagator is a solution. It is found that this requires an additional term. As a by-product new, high precision lattice results for the ghost-gluon vertex in three and four dimensions are obtained.

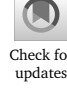

# 1   Introduction

The Gribov-Singer ambiguity [1–4] yields that a perturbatively well-defined gauge fixing prescription, e. g. Landau gauge, becomes ambiguous at the non-perturbative level. This arises as there are many, possibly infinitely many, more solutions to the perturbative gauge condition non-perturbatively. Further constraints cannot be local due to the global nature of the problem, associated with the geometric structure of non-Abelian groups [2,3].

A priori, this does not make perturbative gauges ill-defined. Rather, they act non-perturbatively as a (flat) average over all gauge copies satisfying the perturbative gauge condition [3,5,6]. However, this leads to problems in practical calculations.

One problem is that Gribov copies lead to non-trivial zero modes in the (generalized) Faddeev-Popov operator [1,7,8]. Thus, its inversion in the perturbative gauge-fixing procedure [9] is ill-defined. This has consequences like the breaking of perturbative BRST symmetry [10–12] and negative eigenvalues [13,14] can even alter the spectrum due to cancellations [15]. Another problem is the comparison between lattice calculations and continuum calculations. Even if the previous problem can be solved, lattice calculations usually have operational approaches to fix the gauge by selecting configurations [3]. This is not yet reproducible in continuum calculations. Thus, it is not clear, if continuum results and lattice results for gauge-dependent quantities can be compared at all.

These problems have been attempted to be overcome in a constructive forward approach by various conjectured additional gauge-fixing terms [3,4,16–19]. These prescriptions yield reasonable agreement with lattice results [20–25]. They suffer from the fact that they involve gauge-fixing terms usually not easily implemented in continuum functional calculations, due to involved limiting procedures or non-local terms.

Hence, here an alternative approach will be taken, which essentially amounts to reverse-engineering (part of) the additional gauge-fixing terms in the Lagrangian for a given operational lattice gauge condition, minimal Landau gauge. This approach will be discussed in detail in section 2. To this end, the continuum Dyson-Schwinger equation (DSE) for the ghost will be solved using lattice input, and the difference compared to the ghost propagator from the lattice will be determined. The details for this calculation in SU(2) Yang-Mills theory are described in section 3. Any difference in the thermodynamic limit implies the presence of an additional, ghost-dependent term in the Lagrangian, which could only arise due to gauge-fixing. Indeed, as discussed in section 4, such a difference is found to the extent possible with the employed lattice setups. Studying it for two, three, and four dimensions shows pronounced differences. Especially, in two dimensions the term is qualitatively different, which agrees with the observation that both the lattice and the continuum yield a qualitatively different behavior in two dimensions than in higher dimensions [3,26–29].

The results are summarized in section 5. They suggest that an additional gauge-fixing term is necessary to allow for comparisons between continuum and lattice calculations.

For the present purpose it was necessary to obtain higher precision lattice results for the ghost-gluon vertex in three and four dimensions, which will be presented in appendix A. The input propagators are shown in appendix B for three and four dimensions. For two dimensions these are the same as in [26]

It is important to note that a similar idea was pursued in a forward direction in [30]. There, the DSE for the link on a discrete lattice was derived for the pure Landau gauge condition. Thus, there are additional contributions in the DSE from the finite lattice spacing, which vanish in the thermodynamic limit. Hence, this work has a reverse approach with respect to the present one, where the continuum DSE is used, and the solutions approach their continuum behavior. Nonetheless, similar conclusions are reached.

## 2 Basic idea

A definition of a non-perturbatively well-defined gauge-fixed Euclidean path integral is the one of minimal Landau gauge[1]

$$Z = \int \mathcal{D}A_\mu \Theta\left(-\partial_\mu D_\mu^{ab}\right)\delta\left(\partial_\mu A_\mu^a\right)e^{-\int d^4x \mathcal{L}} \tag{1}$$

$$\mathcal{L} = -\frac{1}{4}F_{\mu\nu}^a F_{\mu\nu}^a$$

$$F_{\mu\nu}^a = \partial_\mu A_\nu^a - \partial_\nu A_\mu^a + gf_{bc}^a A_\mu^b A_\nu^c$$

$$D_\mu^{ab} = \delta^{ab}\partial_\mu + gf_c^{ab}A_\mu^c$$

$$\Theta\left(-\partial_\mu D_\mu^{ab}\right) = \prod_i \theta(\lambda_i), \tag{2}$$

where $-\partial_\mu D_\mu^{ab}$ is the Faddeev-Popov operator and $\mathcal{L}$ is the usual Yang-Mills Lagrangian. This defines minimal Landau gauge, as it is implemented in lattice simulations [3, 31]. It corresponds to a flat average over all Landau gauge Gribov copies with a positive semi-definite Faddeev-Popov operator. This additional restriction can be satisfied by every gauge orbit [32], and many Gribov copies on every gauge orbit do so [33].

In functional calculations it is assumed, based on the fact that the determinant of the Faddeev-Popov operator vanishes on a Gribov horizon [34], that the gauge-fixed Lagrangian inside every Gribov region, and inside every collection of Gribov regions, is identical to the one obtained from the perturbative gauge-fixing,

$$Z = \lim_{\xi \to 0}\int \mathcal{D}A_\mu \mathcal{D}c\mathcal{D}\bar{c}\, e^{-\int d^dx \mathcal{L}_g} \tag{3}$$

$$\mathcal{L}_g = \mathcal{L} + \frac{1}{2\xi}(\partial_\mu A_\mu^a)^2 + \bar{c}_a \partial_\mu D_\mu^{ab}c_b.$$

It has been conjectured that the selection, which Gribov region is actually sampled, should be imposed as a boundary condition to the obtained functional equations [16, 25]. Thus, there should exist some boundary condition for which the obtained correlation functions from (3) coincide with those from (1).

Unfortunately, so far no lattice simulations have been done outside the first Gribov region, as then a sign problem arises. On the other hand, investigations concerning boundary conditions and further restrictions in (1) and functional calculations seem to be consistent with this picture [16, 20–22, 25]. However, there are still problems, which may or may not be due to truncation artifacts in the continuum calculations [16, 25, 29, 35, 36], which leave a nagging doubt.

Therefore, an explicit test of the conjectured equivalence of (1) and (3) will be performed here. To this end, the DSE for the ghost propagator, derived from (3),

$$0 = -D_G^{ab-1}(p) - \tilde{Z}_3 \delta^{ab}p^2 \tag{4}$$
$$+ \int \frac{d^dq}{(2\pi)^d}\Gamma_{0\mu}^{c\bar{c}A,dae}(q,p,-q-p)D_{\mu\nu}^{ef}(p+q)D_G^{dg}(q)\Gamma_\nu^{c\bar{c}A,bgf}(p,q,-p-q),$$

will be used. Herein $D_{\mu\nu}$ is the gluon propagator, $D_G$ is the ghost propagator, $\Gamma_0$ is the tree-level ghost-gluon vertex, and $\Gamma$ is the full ghost-gluon vertex. If this would be the correct DSE from

---

[1]It is implicitly assumed that the so-restricted gauge orbits have, up to a measure-zero set, the same number of gauge copies. If this would not be the case, an orbit-dependent reweighting would be necessary. All available results so far point in the direction that this is not necessary, but to the author's knowledge there is no proof of this yet.

(1) then the correlation functions obtained in lattice simulations from (1) would satisfy (4). If not, then there exists an additional, ghost-dependent term in the gauge-fixed Lagrangian of (1) once ghost fields are introduced. This term is then lacking in the conventional gauge-fixed Lagrangian (3).

Note that this does not necessarily invalidates (3), but only spoils the equivalence between (1) and (3). Still, this would be dramatic enough, as this would forbid to compare lattice results and results from functional methods until a modification of either (1) or (3) reestablishes equivalence. However, this would not invalidate existing results either, as both formulations appear to be valid gauge-fixed formulations. They would just not be in the same gauge.

It should also be kept in mind that even if the DSE is solved, within lattice uncertainties, this does not guarantee the equivalence of (1) and (3), but only does not contradict it. After all, there could be additional terms, which either do not contribute to (4) and/or do not depend on the ghost fields at all, but appear in DSEs of other quantities.

The idea to use lattice inputs to solve continuum DSEs is not new, see e. g. [3,37,38] for reviews. The difference is that all ingredients in (4) are determined using lattice methods, and it is checked, whether the left-hand-side is indeed zero, rather than to leave one element to be determined using the functional equations. In this sense, it is closer to [30] in spirit. It is also different from [30] in the following sense. There, only quantities directly calculated on the lattice enter a DSE for the link. The ghost propagator in (4) is, in contrast to the continuum, not a primary quantity, but rather derived. On the lattice, it is obtained by inverting the Faddeev-Popov operator, which is within the first Gribov region a well-defined procedure [39]. This is consequently also true for the ghost-gluon vertex. See [40] for the particular implementation used to calculate these quantities.

## 3 Setup

### 3.1 Lattice

Table 1: Number and parameters of the configurations used, ordered by dimension, lattice spacing, and physical volume. In all cases $2(10N + 100(d-1))$ thermalization sweeps and $2(N + 10(d-1))$ decorrelation sweeps of mixed updates [40] have been performed, and auto-correlation times of local observables have been monitored to be at or below one sweep. See [41] on details of how the lattice spacing was determined. Config. ghost and gluon are the number of configurations for the ghost propagator and ghost-gluon vertex, and conf. gluon for the gluon propagator, respectively.

| $d$ | $N$ | $\beta$ | $a$ [fm] | $a^{-1}$ [GeV] | L [fm] | config. ghost | config. gluon |
|---|---|---|---|---|---|---|---|
| 3 | 40 | 3.18 | 0.246 | 0.8 | 9.84 | 7645 | 52644 |
| 3 | 60 | 3.18 | 0.246 | 0.8 | 14.8 | 4891 | 42625 |
| 3 | 80 | 3.18 | 0.246 | 0.8 | 19.7 | 3580 | 32805 |
| 3 | 40 | 5.61 | 0.123 | 1.6 | 4.92 | 8220 | 50184 |
| 3 | 60 | 5.61 | 0.123 | 1.6 | 7.38 | 4334 | 38912 |
| 3 | 80 | 5.61 | 0.123 | 1.6 | 9.84 | 2319 | 22563 |
| 3 | 40 | 10.5 | 0.0616 | 3.2 | 2.46 | 8009 | 44021 |
| 3 | 60 | 10.5 | 0.0616 | 3.2 | 3.70 | 5439 | 44072 |
| 3 | 80 | 10.5 | 0.0616 | 3.2 | 4.93 | 1546 | 25371 |
| Continued on next page | | | | | | | |

| Table 1 continued | | | | | | | |
|---|---|---|---|---|---|---|---|
| $d$ | $N$ | $\beta$ | $a$ [fm] | $a^{-1}$ [GeV] | L [fm] | config. ghost | config. gluon |
| 4 | 16 | 2.1306 | 0.246 | 0.8 | 3.94 | 4550 | 40814 |
| 4 | 24 | 2.1306 | 0.246 | 0.8 | 5.90 | 3619 | 31850 |
| 4 | 32 | 2.1306 | 0.246 | 0.8 | 7.87 | 2655 | 15842 |
| 4 | 16 | 2.3936 | 0.123 | 1.6 | 1.97 | 6390 | 42929 |
| 4 | 24 | 2.3936 | 0.123 | 1.6 | 2.95 | 5929 | 45675 |
| 4 | 32 | 2.3936 | 0.123 | 1.6 | 3.94 | 3679 | 1889 |
| 4 | 16 | 2.5977 | 0.0616 | 3.2 | 0.986 | 7150 | 39656 |
| 4 | 24 | 2.5977 | 0.0616 | 3.2 | 1.48 | 3946 | 40562 |
| 4 | 32 | 2.5977 | 0.0616 | 3.2 | 1.96 | 2195 | 19783 |

The DSE (4) requires three central ingredients: The ghost propagator, the gluon propagator, and the ghost-gluon vertex. These have been determined using the methods in [40], for two, three, and four dimensions. For the two-dimensional case the results from [26] have been reused. In three and four dimensions existing data for the ghost-gluon vertex [42] were too noisy, and thus for the present purpose new data was generated. The same goes for the propagators. The corresponding lattice setups and statistics are listed in table 1. The new results for the ghost-gluon vertex are discussed separately in appendix A, and the ones for the propagators are shown in appendix B

## 3.2   Solution of the DSE

The DSE (4) would, in principle, require knowledge of the three input correlation functions at all momenta. On a finite lattice, these are not available, and especially all momenta live on a dual hypercubic lattice. Of course, getting closer and closer to the thermodynamic limit, these problems get less relevant.

To cope with a finite lattice, the following approximations were made:

- The integral was evaluated on a hypercubic lattice, which correspond to the lattice momenta. Momenta appearing in the kernels have been evaluated using the continuum lattice momenta $q_\mu = 2/a \sin(\pi Q_\mu/N)$, where $Q_\mu$ are the integer lattice momenta. The same was done for the external momenta.

- If it was necessary to evaluate propagators on arguments like $(p-q)^2$, which lead to values not mapable to a lattice momentum, the two closest points where used to interpolate.

- The same was done for the ghost-gluon vertex, which is only available for three relative angles [40].

- If the propagators or the ghost-gluon vertex are evaluated at momenta outside the available lattice momenta, the asymptotic form of their dressing functions was used, i. e. zero for the gluon dressing function and its constant value for the ghost dressing function at momenta smaller than the smallest non-zero lattice momentum, and one in all other cases[2].

- The renormalized coupling $g$ was set independently for every lattice setting such that at large momenta, i. e. in the perturbative regime, (4) was solved within statistical errors.

---

[2]This could be improved by using beyond-tree-level perturbation theory at large momenta or extrapolation at small momenta, to reduce lattice artifacts

In addition, the equation was divided by $1/\tilde{Z}_3$ in four dimensions, and this effect was absorbed into the changing value of $g$ and a rescaling of the propagators.

This procedure created the self-energy

$$\Pi(p) = \frac{1}{N_g} \int \frac{d^d q}{p^2 (2\pi)^d} \Gamma_{0\mu}^{c\bar{c}A,dae}(q,p,-q-p) D_{\mu\nu}^{ef}(p+q) D_G^{dg}(q) \Gamma_\nu^{c\bar{c}A,agf}(p,q,-p-q).$$

from the lattice, with $N_g$ the number of gluons. Using also the dressing function $G(p) = p^2 D_G^{aa}(p)/N_g$, reduced (4) to

$$0 = 1 + \frac{1}{G(p)} - \Pi(p).$$

To obtain a statistical error estimate on $\Pi(p)$ it was calculated once with the average values and once with the propagators and vertex modified by its positive or negative 67% bootstrap confidence interval [40]. Likewise, $G(p)$ was obtained from the lattice calculations.

## 4 Results

Allowing now for a misfunction

$$f(p) = 1 + \frac{1}{G(p)} - \Pi(p),$$

this will describe how well the DSE is fulfilled. Of course, on any finite lattice $f(p) \neq 0$, as lattice artifacts will spoil the continuum DSE (4). If these are just lattice artifacts, then $f$ will vanish in the continuum limit. If not, then this implies that (4) is not the appropriate ghost DSE in minimal Landau gauge. Conversely, the calculated ghost dressing function

$$G_c(p) = \frac{1}{1 - f(p) - \Pi(p)} \tag{5}$$

should equal $G(p)$, within statistical errors.

As will be seen, a suitable fit ansatz for $f(p)$ in two dimensions is

$$f(p) = -\frac{A}{p^e} = -\left(\frac{a}{p}\right)^e, \tag{6}$$

while in higher dimensions

$$f(p) = \frac{A - Bp^e}{C + p^{2e}} = r \frac{1 - \left(\frac{p}{b}\right)^e}{1 + \left(\frac{p}{c}\right)^{2e}} \tag{7}$$

was more suitable. Both vanish at large momenta, as befits a feature introduced by Gribov copies. However, the contribution is singular in two dimensions and regular in higher dimensions. Of course, eventually the limit of $a$ and $r$ in the thermodynamic limit will be the most relevant ones, as they determine whether non-zero contributions remain. The values of the determined fit parameters are listed in appendix C.

As is visible in figures 1-3, with a zero misfunction $f$ the lattice propagator indeed does not satisfy the DSE (4) by a substantial amount. Fitting and including a misfunction $f$ using the fit ansätze (6-7), $G_c$ agrees with $G$ within a few percent. This is also shown in figures 1-3. Still, the deviations show substantial differences in the different dimensionalities.

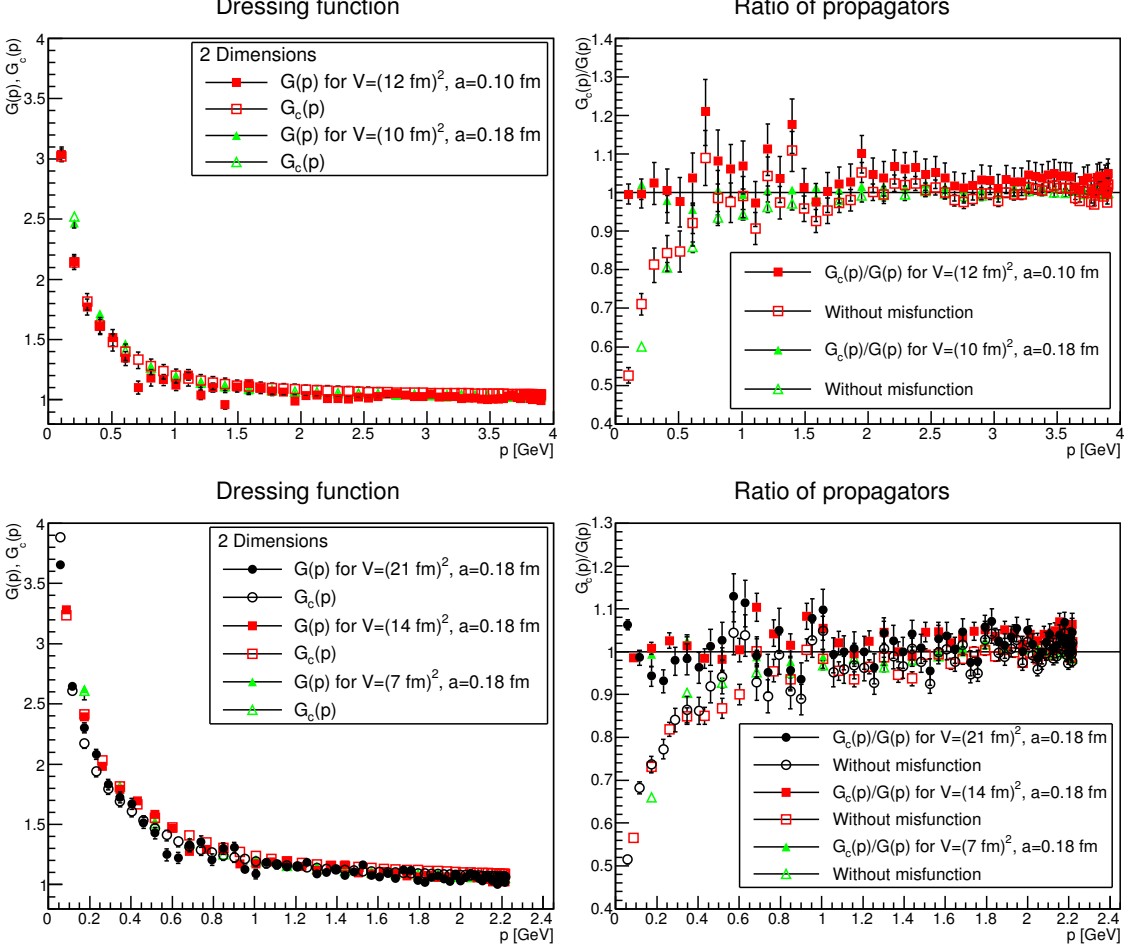

Figure 1: The dressing function in comparison to the DSE result in two dimensions. The left-hand side compares the measured ghost dressing function to the one obtained from (5), i. e. when including the misfunction. The right-hand side shows the ratio of the measured ghost propagator to the calculated one from (5), i. e. with the fitted misfunction, and once without, i. e. $f = 0$. The top panels compare results at different discretizations for roughly fixed physical volume, while the bottom panel compare results at fixed discretizations and different physical volumes.

In two dimensions it appears that a diverging difference is present, starting to become relevant at momenta below roughly half a GeV. At the smallest accessible momenta, this yields already a normalized deviation of 50%. The results only show relatively small dependence on the lattice parameters.

In three dimensions, the difference appears already at about 1 GeV, and shows a pronounced qualitative volume dependence above and below a volume of roughly $(10 \text{ fm})^3$. There is also a significant impact from discretization, which enlarges the effect at larger momenta on finer lattices. Eventually, it seems to be a constant normalized difference in the infrared of about 40%.

In four dimensions, the smaller volumes lead to a much less clearer picture, though again below half a GeV strong, possibly sign-changing effects are observed. Here, larger volumes are clearly necessary. However, all effects set already in at momenta above the smallest non-vanishing one.

The resulting misfunctions are shown in figure 4. Together with the propagator results in

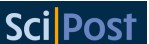

Figure 2: Same as figure 1, but in three dimensions. In addition, the top and middle panels show results for different discretizations at fixed physical volume for two cases.

figure 1-3, these show quite different results in the different dimensionalities.

In two dimensions, the results show that the misfunction is singular, with an exponent of about 0.6 at large volumes. However, the prefactor decreases towards infinite volume, and becomes quite small, but seems to settle at sufficiently large volumes. This is particularly visible in the volume-dependence at fixed momentum. This suggests that even in two dimensions

Figure 3: Same as figure 2, but in four dimensions.

there remains a non-vanishing misfunction, despite the relatively good quantitative agreement between lattice [26] and continuum results [28, 29, 43]. But, as figure 1 shows, this effect is overall small at larger momenta. However, precise calculations of the running coupling in two dimensions [41] indeed support the possibility of a slight discrepancy at very small momenta.

The situation in higher dimensions is quite different. In three dimensions, there is a qualitative change from an infrared positive misfunction to an infrared negative misfunction at a volume of about $(10 \text{ fm})^3$, which is roughly the scale where the maximum of the gluon



Figure 4: The misfunction (left panel) and its value at fixed momentum as a function of volume (right panel) in two dimensions (top panels), three dimensions (middle panels), and four dimensions (bottom panels).

propagator in three dimensions becomes visible [41, 44]. As the momentum dependence of the misfunction shows, this is a behavior which starts at relatively large momenta, and then, with increasing volume, propagates towards the infrared. However, the details depend on the discretization, as a direct comparison at fixed volume in figure 4 shows, starting later at finer discretization, though happening always. Thus, this will require a detailed study of the

thermodynamic limit for a final conclusion.

In four dimensions only comparatively small volumes are available, due to the large computational costs for the ghost-gluon vertex. As so often, a similar behavior as in three dimensions is seen at somewhat smaller volumes [41], especially in the volume dependence at fixed momentum. Also, on the largest volume already a negative contribution propagating towards the infrared is seen, not unlike the situation in intermediate volumes in three dimensions. This again implies the necessity of a more detailed study of the thermodynamic limit.

However, the trend is similar in both three and four dimensions, showing an infrared negative, but constant, misfunction. In all dimensions the results are consistent with an infrared relevant misfunction in the DSE. According to the fits, they behave in two dimensions at large volumes roughly like $\sim p^{-0.6}$ and in three and four dimensions roughly like $(a+p^2)/(b+p^4)$, within large uncertainties of the exponents.

## 5   Summary and discussion

The results presented are, as any lattice result, always improvable with respect to lattice artifacts. Especially finite-volume effects play an important role [3, 41, 45–51]. However, a substantial improvement will be very expensive in terms of computing time. Thus, for the moment, I will take the results at face value, to discuss whether this appears a worthwhile endeavour.

Taken at face value, the results here indicate that the DSE (4) is not the correct equation to describe the ghost propagator in minimal Landau gauge. The most consistent interpretation is that this gauge requires an additional gauge-fixing term, which would modify the DSE. This additional term only contributes at low momenta. This is consistent with the observations in [30].

This would explain why substantial obstacles are frequently encountered when trying to reproduce the characteristic screened lattice results for the propagators using non-perturbative self-consistent solutions in functional methods, see e. g. [25] for an in-depth analysis of this problem. While it is possible to get decent agreement with suitable manipulations, see e. g. [3,35,37,52] for reviews, or suitable adjusted expansion points in semi-perturbative methods, see [4] for a review, a difference in the DSEs would explain the observed problems. It has also been variously argued that additional gauge-fixing terms may be necessary to replicate the lattice minimal Landau gauge [4, 20, 37, 53–58], including various limiting procedures from all Gribov regions [17, 19, 59] or even from the full gauge orbit [18, 23, 24, 60]. Such ideas have been variously implemented approximately in calculations [16, 17, 19, 25, 59], yielding acceptable agreement with lattice results. In fact, there is some lattice evidence which suggests that additional gauge conditions inside the first Gribov region indeed change the propagators, as expected from a gauge choice [21,22]. It could be expected thus that such additional gauge conditions would alter also the DSE, or influence it in some way [16, 20, 61], and thereby produce the different solutions. This is in line with the results obtained in [30] and here.

Alternatively, it has been argued that averaging over all Gribov copies with a flat-weight, a Hirschfeld-Fujikawa gauge [5,6], should yield the ordinary DSEs without modification [39]. It has been argued that such DSEs should yield a so-called [62] scaling behavior [12–14, 16, 63–65], and in fact this type of solution is obtained from them consistently always as well [16, 25, 61, 66, 67].

At the moment, none of the proposed additional gauge-fixings have been possible to be implement simultaneously in lattice calculations and functional calculations. They either suffer from the problem how to implement the cutoff by the functional Θ-function (2) at the first Gribov horizon exactly in the continuum [4, 16, 20], requiring a limiting procedure not yet

replicated in continuum calculations [18, 23], or have a sign problem or coordinate singularity in lattice implementations [5, 6, 12–14, 16, 17, 19, 59, 63–65]. This yields an impasse at the moment for a final resolution of the problem, which requires a full implementation of the same gauge-fixing procedure in both methods. Of course, at energy scales where hadronic physics emerges, this is likely essentially irrelevant, and a comparison in this range appears justified. In fact, once experimental quantities are accessed, they can serve as independent checks of the consistency, and remove the necessity to use the same gauge-fixing procedure in the various methods. This is only an issue as long as gauge-dependent quantities in intermediate steps should be compared. Still, this leaves an unresolved problem.

Being more skeptical of the presented results in this work, many possible improvements are obvious. One is to calculate the ghost-gluon vertex at more momentum configurations, and other measures to improve the numerical solution of the DSE. As always, lattice artifacts are a serious issue. Especially in four dimensions, but also in lower dimensions, substantially larger volumes along lines-of-constant physics need to be accessed. Only if pushed into the far asymptotic regime, this could really be taken literally. But while very expensive in terms of computing time, this is straightforward.

At a second level of improvement, it would be good to redo the same exercise with the gluon equation or other DSEs[3]. However, because of the two-loop terms and the appearing four-point functions, this will be technically at least two orders of magnitude more demanding [35]. To support that the observed effect is a gauge artifact, it would be useful to repeat the same study with other ways of completing the Landau gauge [3, 17, 20–22] in the first Gribov region. If the present interpretation is correct, this should change the misfunction. However, this will be even more expensive.

Still, this would be worthwhile, as this path could finally decide whether the infrared of gauge-dependent Yang-Mills correlation functions is indeed determined by gauge-fixing. Also, it would establish a new level of trust to comparison of gauge-dependent results between different methods.

# Acknowledgments

I am grateful to Reinhard Alkofer, Markus Huber, and André Sternbeck for a critical reading of the manuscript and valuable feedback.

# A  Ghost-gluon vertex

The present investigation required an update[4] in precision in the ghost-gluon vertex in three and four dimensions. Following [40], the form factor was calculated in three different kinematic configurations. In the back-to-back configuration the gluon has zero momentum. In the symmetric one all three momenta have equal magnitude. In the last one, the ghost momentum and the gluon momentum are orthogonal, which has the largest integration measure in the angular integral in the DSE (4), but otherwise unconstrained momenta. Technical details can be found in [40].

The results are shown in figure 5 for three dimensions and in figure 6 for four dimensions. The results are consistent with previous investigations for the cases with non-zero gluon mo-

---

[3]It should be possible to do this also for FRGs, where the regulator can be included by reweighting, in a similar manner as sources [68].

[4]Note that slightly higher statistics than listed in table 1 have been used to create the figures in the appendices.

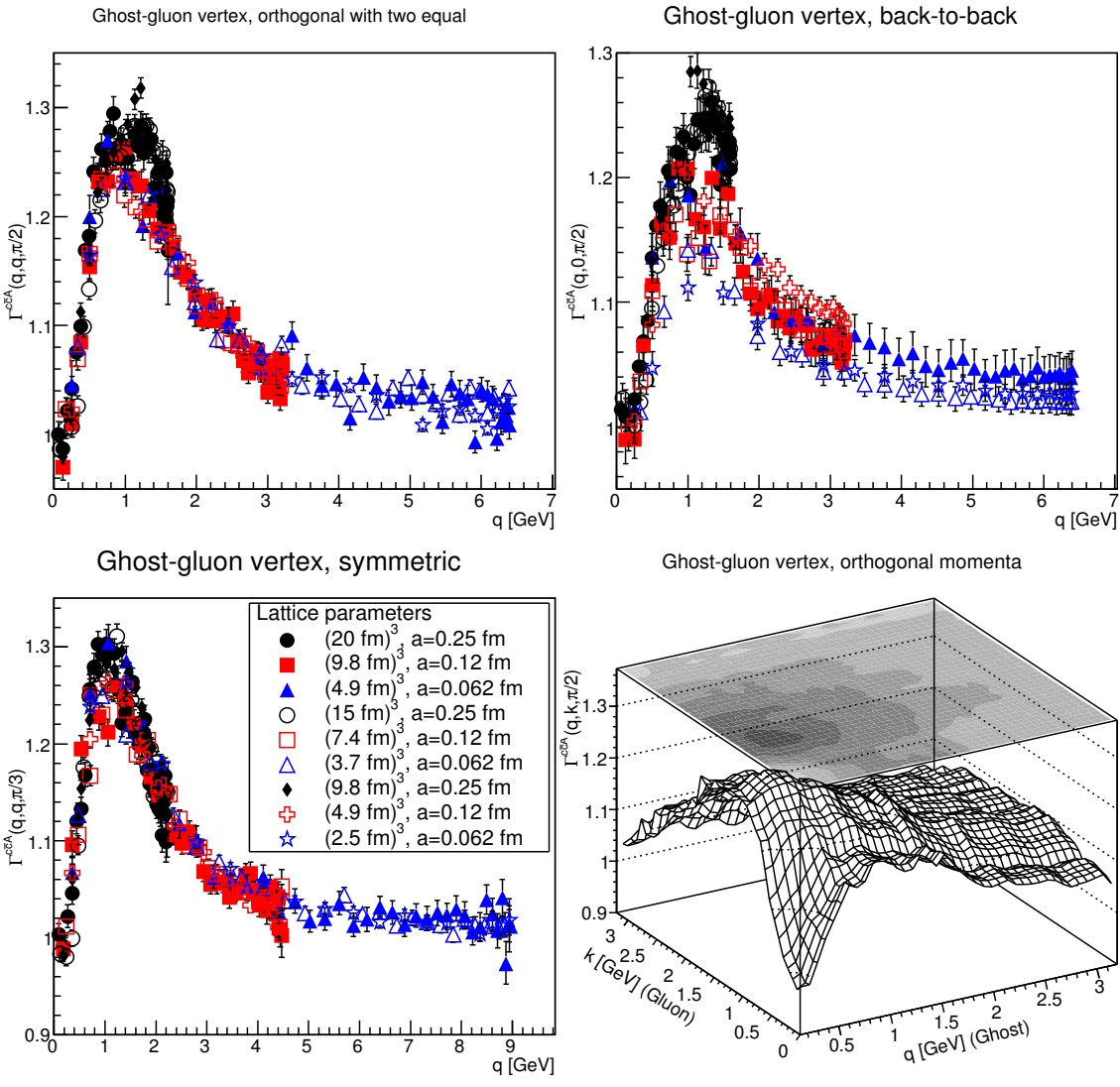

Figure 5: The ghost-gluon vertex in three dimensions. The top-left panel shows a cut along the diagonal of the lower-right plot. The latter shows the situation with the gluon momentum being orthogonal to the ghost momentum, with interpolation of the data points for the largest volume at the intermediate discretization. The top-right panel shows the back-to-back momentum configuration and the bottom-left panel the symmetric momentum configuration.

mentum [40, 42]. The deviations from tree-level are stronger in four dimensions, and the vertex is consistent with one at vanishing (anti-)ghost momentum. Overall, the deviation from tree-level is small.

However, the situation at vanishing gluon momentum shows the effect of so far not yet observed discretization artifacts. It is visible that at coarse discretization the ghost-gluon vertex deviates more strongly from the tree-level form than at finer discretizations at fixed momentum. Especially, the distinct maximum at non-zero gluon momentum seems to be substantially reduced on finer lattices at zero gluon momentum. In four dimensions this effect is somewhat less pronounced. Thus, the angular dependence is not trivial.

This dependence on lattice parameters is emphasized in figures 7 and 8. It shows , especially in three dimensions, but also in four, that when moving from the coarsest to the next finer discretization the peak height reduces in the back-to-back configuration. However, there

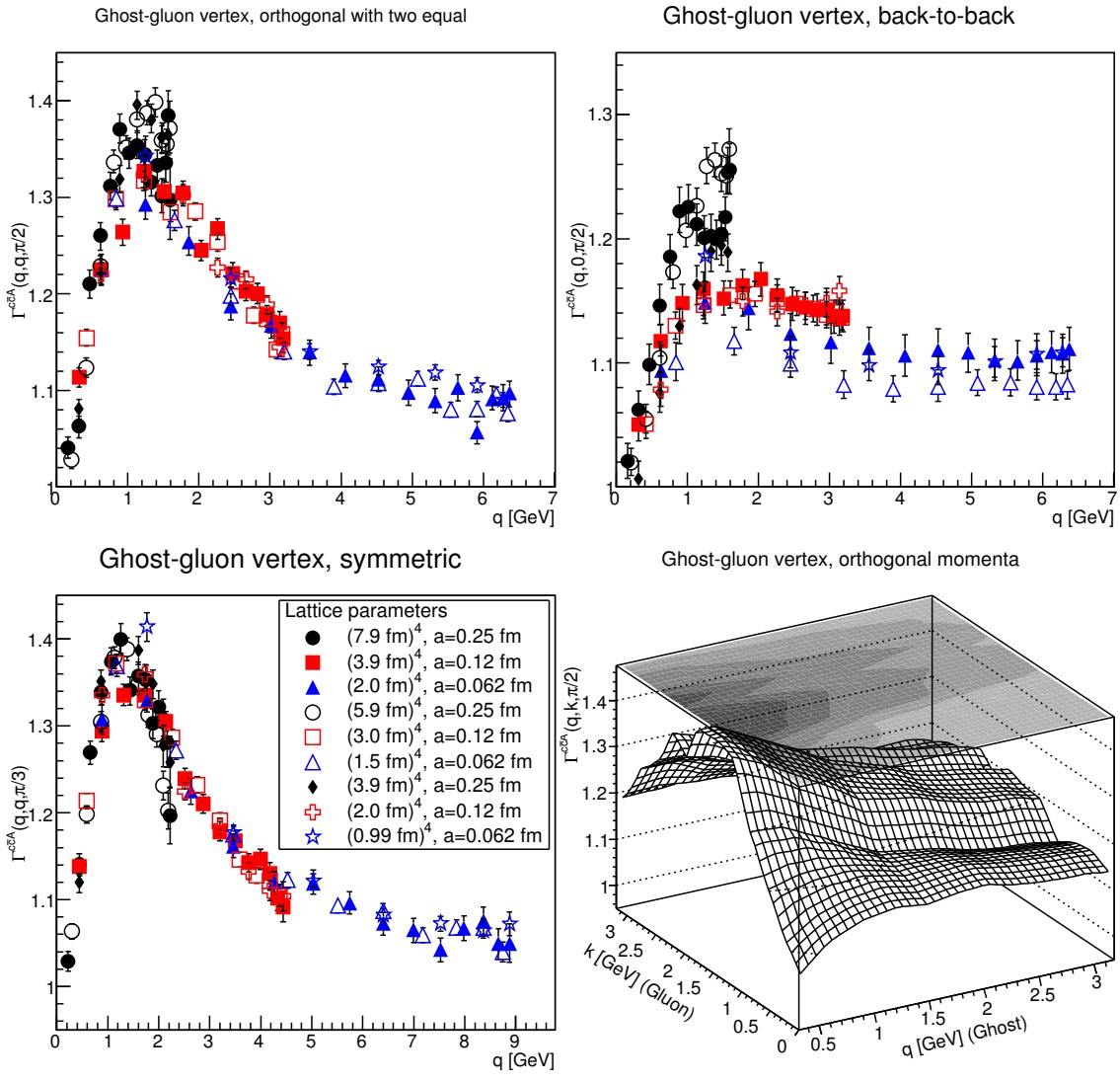

Figure 6: The ghost-gluon vertex in four dimensions. The top-left panel shows a cut along the diagonal of the lower-right plot. The latter shows the situation with the gluon momentum being orthogonal to the ghost momentum, with interpolation of the data points for the largest volume at the intermediate discretization. The top-right panel shows the back-to-back momentum configuration and the bottom-left panel the symmetric momentum configuration.

is little difference when moving then to the finest discretization. No similar effect is seen in the symmetric configuration. As there no momentum vanishes, which is more sensitive to a finite volume, this is not completely surprising. This suggests, in line with other results from propagators [41, 45–51], that a discretization above or around $(1.5\,\text{GeV})^{-1}$ appears necessary to also assess finite-volume effects quantitatively reliable.

## B  Propagators

The results for the propagators involved are shown in figure 9 and figure 10 in three and four dimensions, respectively. The results for two dimensions are the same as in [26].

Figure 7: The systematic dependence on the lattice parameters for the ghost-gluon vertex in three dimensions. The left panels show the symmetric momentum configuration and the right panels the back-to-back momentum configuration. The top and middle panels show two cases with fixed physical volume and changing discretization and the bottom panel a case at fixed discretization and changing physical volume.

It is visible that, especially in four dimensions, substantial lattice artifacts are present, especially in the infrared finite-volume effects. However, larger volumes are currently not accessible due to the requirements of computation time for the necessary statistics for the ghost-gluon vertex. Still, the largest volume in four dimension, and several lattice settings

Figure 8: The systematic dependence on the lattice parameters for the ghost-gluon vertex in four dimensions. The left panels show the symmetric momentum configuration and the right panels the back-to-back momentum configuration. The top and middle panels show two cases with fixed physical volume and changing discretization and the bottom panel a case at fixed discretization and changing physical volume.

in three dimensions, approach the qualitative asymptotic behavior, which has been seen in many lattice investigations [41, 45–51]. Only the finite limit at zero momentum of the ghost dressing function in the far infrared is not yet manifest in the data. However, this is mimicked by setting it to a finite value, which is though somewhat low, when using it below this value,



Figure 9: The ghost dressing function and the gluon propagator for the different lattice settings in three dimensions. Lattice volume increases from left to right and the latte spacing decreases from top to bottom. Results are along all possibles axes [40]. Errors are smaller than the symbol sizes, and the same number of configurations have been used in both cases.

as described in section 3.2.

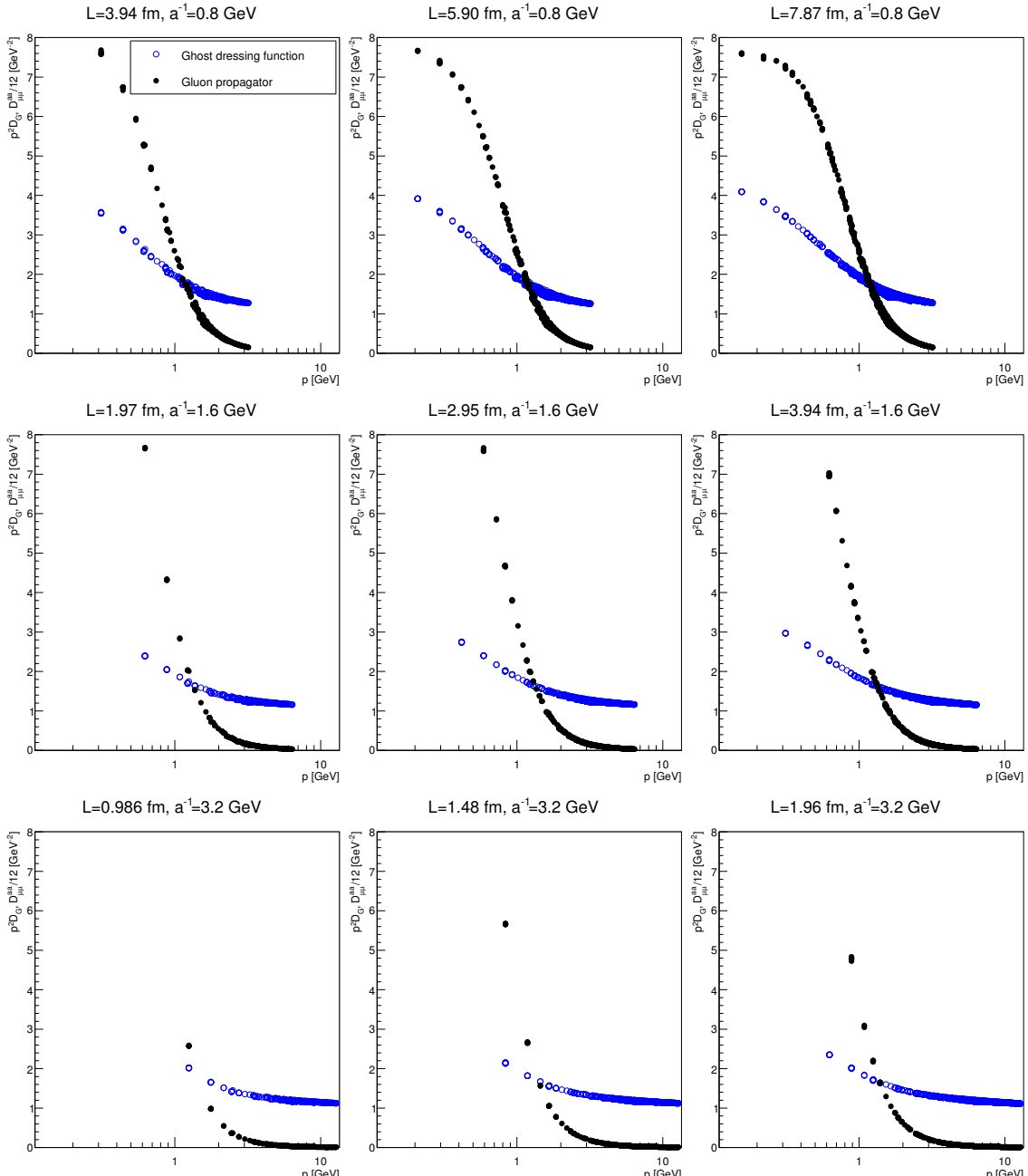

Figure 10: The ghost dressing function and the gluon propagator for the different lattice settings in four dimensions. Lattice volume increases from left to right and the latte spacing decreases from top to bottom. Results are along all possibles axes [40]. Errors are smaller than the symbol sizes, and the same number of configurations have been used in both cases. The results have not been renormalized,

# C   Fit parameters

Table 2: The fit parameters of the fits (6) and (7) in two, three, and four dimensions, respectively. Note that no dimensions are given for $A$, $B$, and $C$, as to suppress error propagation due to $e$, the propagators have been made initially dimensionless by rescaling with 1 GeV. If desired, the results can be reconverted in a straightforward way.

| $d$ | $N$ | $a$ [fm] | $e$ | $A$ | $B$ | $C$ |
|---|---|---|---|---|---|---|
| 2 | 120 | 0.10 | $0.7^{+1.4}_{-0.1}$ | 0.07(6) | | |
| 2 | 100 | 0.10 | 0.7(1) | 0.08(4) | | |
| 2 | 80 | 0.10 | 0.78(1) | 0.07(2) | | |
| 2 | 60 | 0.10 | 1.1(1) | 0.05(1) | | |
| 2 | 40 | 0.10 | 1.2(1) | 0.077(6) | | |
| 2 | 20 | 0.10 | 1.8(2) | 0.13(1) | | |
| 2 | 120 | 0.18 | 0.7(5) | $0.04^{+0.08}_{-0.01}$ | | |
| 2 | 100 | 0.18 | $0.6^{+1.1}_{-0.2}$ | $0.04^{+0.04}_{-0.03}$ | | |
| 2 | 80 | 0.18 | 0.6(1) | 0.06(3) | | |
| 2 | 60 | 0.18 | 0.62(2) | 0.07(3) | | |
| 2 | 40 | 0.18 | 1.4(4) | 0.02(1) | | |
| 2 | 20 | 0.18 | 1.5(1) | 0.055(3) | | |
| 3 | 80 | 0.062 | 2.2(2) | 0.0040(6) | 0.036(3) | 0.036(2) |
| 3 | 60 | 0.062 | 1.9(1) | 0.11(2) | 0.17(1) | 0.37(2) |
| 3 | 40 | 0.062 | 1.9(1) | 0.64(2) | 0.31(1) | 3.4(3) |
| 3 | 80 | 0.12 | 2.0(1) | -0.006(3) | 0.042(2) | 0.07(2) |
| 3 | 60 | 0.12 | 1.6(1) | 0.065(9) | 0.11(2) | 0.19(2) |
| 3 | 40 | 0.12 | 1.7(1) | 0.26(2) | 0.25(2) | 0.99(5) |
| 3 | 80 | 0.25 | 2.0(4) | -0.26(5) | -0.17(4) | 1.9(1) |
| 3 | 60 | 0.25 | 1.7(1) | -0.07(4) | 0.12(3) | 0.6(2) |
| 3 | 40 | 0.25 | 1.7(1) | 0.15(3) | 0.23(3) | 0.67(6) |
| 4 | 32 | 0.062 | 2.0(1) | 0.077(2) | 0.053(2) | 0.13(1) |
| 4 | 24 | 0.062 | 2.3(1) | 1.0(2) | 0.35(1) | 2.5(1) |
| 4 | 16 | 0.062 | 4.0(1) | 433(5) | 6.9(2) | 3755(14) |
| 4 | 32 | 0.12 | 1.7(1) | 0.036(1) | 0.061(2) | 0.073(1) |
| 4 | 24 | 0.12 | 1.9(1) | 0.38(1) | 0.22(1) | 0.49(1) |
| 4 | 16 | 0.12 | 1.8(1) | 3.3(1) | 0.63(1) | 6.2(1) |
| 4 | 32 | 0.25 | 1.7(2) | 0.0075(3) | 0.030(2) | 0.038(1) |
| 4 | 24 | 0.25 | 1.9(1) | 0.14(1) | 0.19(1) | 0.25(1) |
| 4 | 16 | 0.25 | 1.7(1) | 1.7(1) | 0.46(1) | 1.8(1) |

The results for the fits (6) and (7) are listed in table 2. It should be noted that the fit parameters are correlated. Thus, varying the parameters of the fits freely inside the error bands given in table 2 creates worse agreement with the data then when taking the correlations into account. This is done for the errors in figure 4.

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
