# Peer review of "Constraining the gauge-fixed Lagrangian in minimal Landau gauge"

_SciPost Physics, doi:SciPost Phys. 8, 071 (2020)_

## Round 1 · Referee Report · Anonymous (Referee 1) · 2019-12-26

Strengths

1- Well written 2- Problem addressed clearly stated 3- Study how to compare different non-perturbative techniques results for QCD 4- New way to address the comparison

Weaknesses

1- Does not include a full discussion of all issues 2- Biased discussion at the end

Report

Axel Maas
"Constraining the gauge-fixed Lagrangian in minimal Landau gauge"
arXiv:1907.10435

The manuscript under appreciation discuss an important issue for the non-perturbative regime of QCD, the comparison between lattice simulations and Dyson-Schwinger (DS) results. The manuscript is well written and the problem is clearly stated. After a general introduction, the author writes the generating functional for the continuum formulation of the minimal Landau gauge, that is “identified” with the continuum version of the lattice formulation, and the usual Faddeev-Popov (FP) functional. Then the FP is used to write the Dyson-Schwinger equation for the ghost propagator that involves the gluon and ghost propagators and ghost-gluon vertex. This is one of the simplest DS equations and the author uses it as a benchmark to compare lattice and DS results.

(i) Although the author provides a good discussion of various issues, he forgets to mention that on the lattice, if one excludes the corresponding perturbative treatment, there are no ghost fields and the ghost propagator is “recovered” mimicking the usual continuum procedure. To the referee this is an important point that the author should mention in Sec. 2 to alert the reader.

The results of 3D and 4D lattice simulations, for various lattice spacings and volumes, are used in the ghost DS equation to check if the equations is solved exactly. The full exercise requires the computation of the gluon and ghost propagators, together with the lattice computation of the ghost-gluon vertex that is performed for three kinematical configurations. In all the computations large statistical ensembles are used. The manuscript includes a nice discussion of the results obtained for the various Green functions and of the deviations between lattice and continuous calculations.

(ii) However, the manuscript does not discuss in detail the effects of discretisation effects on the ghost-gluon vertex and on the propagators. In particular, the propagators are not shown at all and they should be reported. This is of at most importance as it is well known [O. Oliveira, P. J. Silva, Phys. Rev. D86 (2012) 114513] that for the propagators the lattice spacing and the lattice volume should be properly chosen. A possible explanation of the observed differences could be due, at least partially, for the choice of lattice spacings and volumes where the simulations are performed

(iii) The finite volume effects, being due to the lattice spacing and to the lattice volume, are not discussed properly. This is also important as the number of lattice spacings and volumes reported on the manuscript varies significantly. For example, looking at the 3D simulations with lattice sizes of ~2 fm or ~ 3.9 fm, or 4D simulations with lattice sizes ~4.9 fm or ~9.8 fm, one can see clear differences between the various data sets for the various kinematical configurations. The grouping together of all the simulations data on Figs. 5 and 6 only makes their reading quite difficult. My suggestion goes to add the results for the same volumes separately. This can be done either as an inset on the figs. or by adding extra plots.

(iv) The section that worries me most being the Summary. I completely disagree with the authors summary. First, given the finite volume effects I hardly agree with what is written there. At most, would say that this study suggests that the comparison should be done with care and further studies are need before doing any conclusion. I cannot agree, in any way, with the second paragraph of the Summary. I recall the author that there are other interpretations [M. Tissier, Phys. Lett. B784 (2018) 146] and although the efforts associated with the scaling solution are an important work and contribution to the understanding of QCD, there is no evidence of these type of solution coming from lattice [probably the closest being A. Sternbeck, M. Mueller-Preuskker, Phys. Lett. B 726 (2013) 396].

The author should include all the above mentioned references in the revision.

In summary, I do think that the manuscript contains new and important contributions that can help in the understanding of QCD. It certainly, deserves to be published but I cannot agree with the present form of the manuscript and ask the author to review it.

Requested changes

See the report.

  • validity: -
  • significance: -
  • originality: -
  • clarity: -
  • formatting: -
  • grammar: -

Author:  Axel Maas  on 2020-01-27  [id 721]

(in reply to Report 1 on 2019-12-26)
Category:
answer to question
correction

Dear referee,

I am grateful for the constructive referee report, and have made the following changes:

(i) Although the author provides a good discussion of various issues, he forgets to mention that on the lattice, if one excludes the corresponding perturbative treatment, there are no ghost fields and the > ghost propagator is “recovered” mimicking the usual continuum procedure. To the referee this is an important point that the author should mention in Sec. 2 to alert the reader.

I have added this in the last paragraph of section 2.

(ii) However, the manuscript does not discuss in detail the effects of discretisation effects on the ghost-gluon vertex and on the propagators. In particular, the propagators are not shown at all and they > should be reported. This is of at most importance as it is well known [O. Oliveira, P. J. Silva, Phys. Rev. D86 (2012) 114513] that for the propagators the lattice spacing and the lattice volume should be > properly chosen. A possible explanation of the observed differences could be due, at least partially, for the choice of lattice spacings and volumes where the simulations are performed

I have added another appendix with the results for the propagators. I have also extended the discussion concerning systematic artifacts.

(iii) The finite volume effects, being due to the lattice spacing and to the lattice volume, are not discussed properly. This is also important as the number of lattice spacings and volumes reported on the > manuscript varies significantly. For example, looking at the 3D simulations with lattice sizes of ~2 fm or ~ 3.9 fm, or 4D simulations with lattice sizes ~4.9 fm or ~9.8 fm, one can see clear differences > > between the various data sets for the various kinematical configurations. The grouping together of all the simulations data on Figs. 5 and 6 only makes their reading quite difficult. My suggestion goes to > add the results for the same volumes separately. This can be done either as an inset on the figs. or by adding extra plots.

I have added corresponding plots in the style of the main text figures 1-3 to compare fixed discretization or volume for two particular momentum configurations. I have also extended the discussion concerning artifacts in the whole manuscript.

(iv) The section that worries me most being the Summary. I completely disagree with the authors summary. First, given the finite volume effects I hardly agree with what is written there. At most, would say > that this study suggests that the comparison should be done with care and further studies are need before doing any conclusion.

I have toned down the corresponding statement, and emphasized the need for more systematics.

I cannot agree, in any way, with the second paragraph of the Summary. I recall the author that there are other interpretations [M. Tissier, Phys. Lett. B784 (2018) 146] and although the efforts associated with the scaling solution are an important work and contribution to the > understanding of QCD, there is no evidence of these type of solution coming from lattice [probably the closest being A. Sternbeck, M. Mueller-Preuskker, Phys. Lett. B 726 (2013) 396].

I have rewritten the entire second paragraph, and quite a bit of the whole summary, completely. All mention of the scaling solution are now clearly marked as speculation in situations not addressable in current lattice calculations, and the current situation, that a screened behavior in all current lattice calculations (in 3d and 4d) is unambiguously established, is fully appreciated. However, until we are able to do the same calculations in the continuum and the lattice, and especially the Hirschfeld-Fujikawa-type-gauge has been done on the lattice, I cannot agree that this is already fully understood. Unfortunately, we still do not have yet any gauge-fixing procedure, which resolves the Gribov-Singer ambiguity, and is simultaneously practically implementable both in continuum calculation and on the lattice, at least as far as I currently can see. E.g. the one of Tissier you mention has a sign problem (due to the factors s(i), and also the counting of unstable directions is not trivial) on the lattice (which is also true for the Hirschfeld-Fujikawa gauge).

I still have considerably extended the whole discussions, and hopefully have made clearly distinct what is pretty certain - the situation in minimal Landau gauge on the lattice and the existence of screened solutions to the functional equations in truncations - and those which is speculative - everything concerning the scaling solution and what happens in unmodified Hirschfeld-Fujikawa gauge. In particular that no lattice calculation provides any indication of a scaling solution and that such a result is also not expected in any kind of lattice simulations inside the first Gribov region. Still, I think the possibilities beyond that what is currently possible on the lattice deserves mention.

I apologize if the original version created the impression that this issue was settled.

As requested, I have added the provided references, alongside several further ones.

I have also found an error in figures 2 and 3 with respect to the compared lattice parameters, and fixed it. I also added a third comparison, to highlight even further systematic effects.

---

## Round 2 · Referee Report · Anonymous · 2020-2-18

Strengths

1. A new view of an old an important problem concerning the comparison of the results from different non-perturbative techniques for gauge dependent Green's functions
2. It raises a number of questions and open a new direction of research.

Weaknesses

1. Limited access to physical volumes that raise many questions on the authors conclusions.

Report

The author have adequately responded to all issues raised in the report and included all my suggestions. He also produced a new version of section 5 - "Summary and discussion" that greatly improve the reading of the paper contribution to the discussion.

Requested changes

No changes.

---

## Editorial Decision

published